# VISUAL SPATIAL TUNING

## ABSTRACT

Capturing spatial relationships from visual inputs is a cornerstone of human-like general intelligence. Several previous studies have tried to enhance the spatial awareness of Vision-Language Models (VLMs) by adding extra expert encoders, which brings extra overhead and usually harms general capabilities. To enhance the spatial ability in general architectures, we introduce Visual Spatial Tuning (VST), a comprehensive framework to cultivate VLMs with human-like visuospatial abilities, from spatial perception to reasoning. We first try to enhance spatial perception in VLMs by constructing a large-scale dataset termed VST-P, which includes 4.1M samples spanning 19 skills across single view, multiple images, and videos. Then, we present VST-R, a curated dataset with 135K samples that instruct models to reason in space. In particular, we adopt a progressive training pipeline: supervised fine-tuning to build foundational spatial knowledge, followed by reinforcement learning to improve spatial reasoning abilities further. Without the side-effect to general capabilities, the proposed VST consistently achieves state-of-the-art results on several spatial benchmarks, including 34.8% on MMSI-Bench and 61.2% on VSI-Bench. It turns out that the Vision-Language-Action models can be significantly enhanced with the proposed spatial tuning paradigm, paving the way for more physically grounded AI.

## 1 INTRODUCTION

Vision-Language Models (VLMs) (Achiam et al., 2023; Comanici et al., 2025; Guo et al., 2025b; Wang et al., 2024b; Chen et al., 2024c) have achieved remarkable success across a wide range of domains, such as visual question answering (Yue et al., 2024; Liu et al., 2024c), document understanding (Liu et al., 2024d; Mathew et al., 2021), and autonomous GUI agents (Xie et al., 2024). However, these models exhibit limitations in capturing spatial relationships from sequential visual observations (Yang et al., 2025c;a). This spatial understanding ability is a foundational component of general intelligence, presents across a broad spectrum of animals, including humans (Hegarty et al., 2006; Piaget, 2013). The deficiency significantly constrains current VLMs to effectively interact with the physical world, thereby limiting their application in fields such as robotics (Brohan et al., 2022; Zitkovich et al., 2023), autonomous driving (Tian et al., 2024), and augmented/virtual reality (AR/VR) (Grauman et al., 2022). To mitigate this issue, several studies have explored the incorporation of additional 3D-aware expert encoders (Fan et al., 2025; Bigverdi et al., 2025). However, this approach often introduces extra complexity and can negatively impact the general capabilities of the models. Alternatively, other research efforts have focused on the development of specialized datasets (Chen et al., 2024a; Daxberger et al., 2025; Zhang et al., 2025a; Xu et al., 2025b; Yin et al., 2025; Ouyang et al., 2025), aiming to enhance the spatial understanding abilities of VLMs.

Nevertheless, these arts have typically concentrated on limited or isolated aspects of spatial understanding. As summarized in Table 1, some studies focus only on the supervised fine-tuning stage, while others are restricted to the single scenario, overlooking the diversity of visual input. To this end, we introduce a comprehensive and integrated framework, termed Visual Spatial Tuning (VST), which is designed to cultivate human-like visuospatial abilities in VLMs holistically. As illustrated in Figure 1, VST effectively augments the spatial capabilities of existing VLMs through the construction of an extensive and carefully curated dataset. This enhancement proves advantageous for downstream Vision-Language-Action (VLA) tasks.

To develop the VST, we deconstruct spatial ability into two key components: *spatial perception* and *spatial reasoning*. We define spatial perception as the ability to discern the spatial relationships

| Method | Data Type | | | Data Usage | |
|---|---|---|---|---|---|
| | SI | MI | Video | SFT | RL |
| SpatialVLM | ✓ | ✗ | ✗ | ✓ | ✗ |
| SAT | ✓ | ✗ | ✗ | ✓ | ✗ |
| MM-Spatial | ✓ | ✗ | ✗ | ✓ | ✗ |
| SPAR | ✓ | ✓ | ✓ | ✓ | ✗ |
| Space-R | ✗ | ✗ | ✓ | ✓ | ✓ |
| VLM-3R | ✗ | ✗ | ✓ | ✓ | ✗ |
| **VST (ours)** | ✓ | ✓ | ✓ | ✓ | ✓ |

Table 1: Spatial dataset comparison.

Figure 1: Overview of our VST framework.

between objects, and spatial reasoning as the ability to build and mentally manipulate an internal model of an environment. These two components correspond to the concepts of perceptual and conceptual spatial ability, respectively, as proposed in cognitive science (Piaget, 2013). Effective spatial perception requires the model to possess foundational spatial knowledge—specifically, the ability to identify both "what is it?" and "where is it?" within its peripersonal space. While existing VLMs can accurately recognize objects and locate them within pixel space using 2D points or bounding boxes (Wang et al., 2024b; Bai et al., 2025; Deitke et al., 2025), their ability to determine object positions in 3D space remains limited (Ma et al., 2024; Tong et al., 2024). Therefore, we introduce the VST-Perception (VST-P) dataset, comprising 4.1 million samples across 19 diverse tasks. It incorporates single-image data to facilitate VLMs in discerning spatial relationships beyond the pixel level, which is an essential step towards bridging the gap between pixel space and 3D space. In addition, multi-image data is included to enhance the ability to comprehend spatial relationships from multiple viewpoints, and video data enables the capture of spatiotemporal relationships. Collectively, this dataset provides a comprehensive foundation for advancing spatial perception in VLMs.

Beyond foundational spatial perception, we expect the model to mentally represent spatial relationships beyond its own body, thereby engaging in advanced spatial reasoning. To this end, we introduce the VST-Reasoning (VST-R) dataset, which comprises samples featuring chain-of-thought (CoT) processes to facilitate the spatial reasoning ability, as well as samples with rule-checkable answers to further enhance its reasoning capabilities. In spatial reasoning, we place particular emphasis on multi-image scenarios, as these necessitate the model's ability to identify connections among objects and cameras, and to mentally reconstruct spatial layouts. However, when generating spatial CoT, the limited multi-view spatial understanding of current large VLMs (Yang et al., 2025a;c) poses challenges for directly synthesizing accurate layout descriptions and coherent reasoning chains. Drawing inspiration from human cognition, we propose prompting with Bird's-Eye View (BEV) annotation. It leverages a top-down perspective to explicitly convey spatial relationships between objects, thereby improving the quality of both generated layout descriptions and reasoning process.

Building upon the introduced VST-P and VST-R datasets, we propose to inject visual spatial knowledge into VLMs through supervised fine-tuning and further enhance spatial reasoning capabilities via reinforcement learning (RL). This progressive approach mirrors the development of human spatial intelligence (Piaget, 2013), i.e., establishing a foundation in spatial perception before developing higher-level spatial reasoning abilities. As a result, the VST framework consistently achieves state-of-the-art performance on multiple spatial benchmarks, attaining 86.5% on CV-Bench (Tong et al., 2024), 34.8% on MMSI-Bench (Yang et al., 2025c), and 61.2% on VSI-Bench (Yang et al., 2025a), while preserving the general multi-modal capabilities. Furthermore, the spatial proficiency acquired via VST demonstrably enhances broader VLA tasks. For instance, Qwen2.5VL-3B (Bai et al., 2025) fine-tuned on our VST yields an 8.6% improvement on the LIBERO benchmark (Liu et al., 2023).

## 2 DATASET

In this section, we introduce the VST dataset, specifically developed to enhance the spatial perception and reasoning capabilities of VLMs. First, we construct a large-scale dataset, **VST-Perception (VST-P)**, to equip VLMs with comprehensive spatial knowledge. Building upon this foundation, we further create the **VST-Reasoning (VST-R)** dataset to enable VLMs to reason in space.

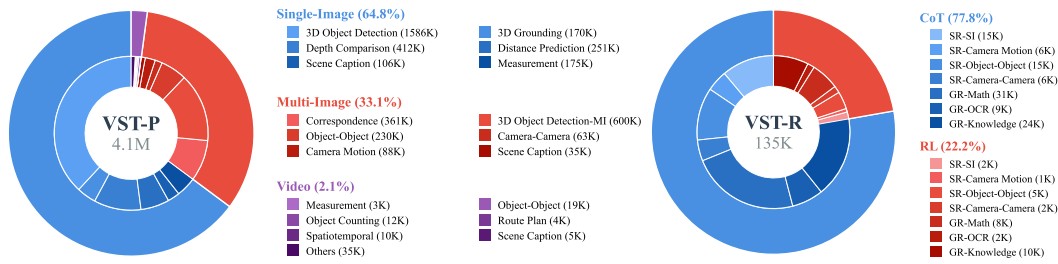

(a) Perception data distribution.    (b) Reasoning data distribution

Figure 2: Overview of the VST dataset. (a) The distribution of VST-P, which is used for SFT. (b) The distribution of VST-R, which is used for CoT cold start and RL. 'SR' denotes spatial reasoning, and 'GR' denotes general reasoning.

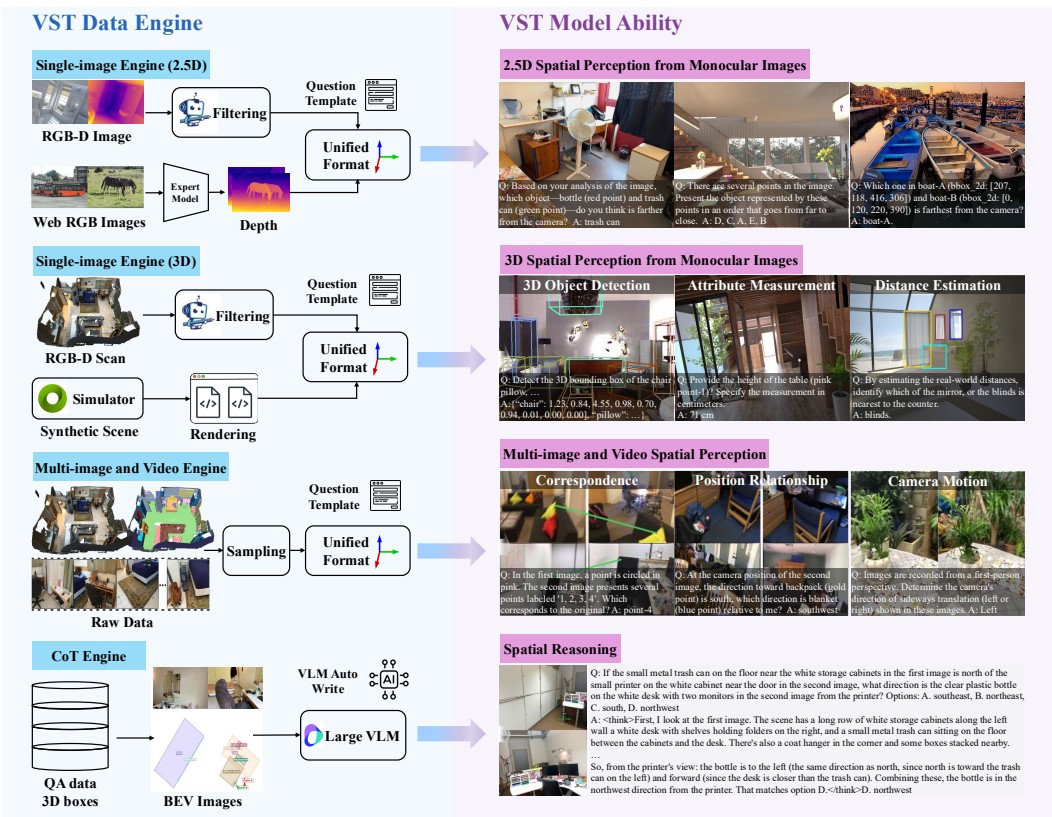

Figure 3: Data engines of VST (left) and the capabilities they enable in VST-Model (right).

## 2.1 VST-PERCEPTION

As illustrated in Figure 2a, the VST-P dataset contains **4.1 M** samples across **19** different tasks for supervised fine-tuning, covering three primary vision scenarios, i.e., single-image, multi-image, and video. Specifically, single-image data constitutes the majority (64.8%), multi-image data accounts for 33.1%, and video data makes up the remaining small portion (2.1%).

**Single-image.** Since monocular images are easily obtainable, single-image data constitutes the largest category. This category primarily encompasses tasks such as relative depth estimation (2.5D), 3D object detection, and distance estimation. These tasks bridge the gap between 2D pixel coordinates and the 3D physical world, thereby facilitating the acquisition of spatial knowledge and the development of spatial awareness in VLMs. To collect this data, we create dedicated data engines to gather data with depth maps and data with 3D bounding box annotations, as shown in the top left of Figure 3. The depth data mainly comes from public datasets and synthetic data. The open-source data originates from ScanNet++ (Yeshwanth et al., 2023), which is collected using real-world devices, and Hypersim (Roberts et al., 2021), which is generated by a simulator. To increase the diversity of

depth data, we use a depth expert model (Yang et al., 2024b) to create pseudo labels for wild images from the COCO dataset (Lin et al., 2014). After obtaining the depth maps, we convert them to the same coordinate system and generate depth-related visual instruction samples. The reference formats for depth-related samples encompass text-, point-, box-, and visual-prompt-based representations. These diverse formats enable VLMs to infer the relative distance from objects to the camera plane.

For the 3D data engine, we use two main approaches. The first approach relies on open-source datasets, including ScanNet (Dai et al., 2017), ARKitScenes (Baruch et al., 2021), Hypersim (Roberts et al., 2021), SUN-RGBD (Song et al., 2015), Matterport3D (Chang et al., 2017), and Objectron (Ahmadyan et al., 2021). Since the 3D bounding boxes from ScanNet and Matterport3D are axis-aligned, we use the corrected versions from EmbodiedScan (Wang et al., 2024c) to ensure greater accuracy. Notably, each dataset is designed for distinct applications, provides visual data in either video or image format, and annotates objects in varying coordinate systems. Therefore, we standardize all collected 3D bounding boxes to a unified camera coordinate system and process the raw visual data to reduce repetition and occlusion. The second approach is generating data using a simulator. Specifically, we use Isaac Sim to synthesize data, and scenes are from the GUTopia (Wang et al., 2024a). With the large-scale data with 3D bounding boxes, we create visual instruction samples for 3D object detection, 3D grounding, attribute measurement, and distance estimation tasks.

In the 3D object detection task, we predict the 9-DoF bounding box in the camera coordinate system. Specifically, the 3D bounding box is defined by $(x, y, z, x_l, y_l, z_l, p, y, r)$, where $(x, y, z)$ are the box center, $(x_l, y_l, z_l)$ are the length along the X, Y, and Z axes, and $(p, y, r)$ are the rotation angles. However, a significant challenge in utilizing datasets aggregated from disparate sources is the inherent variability in camera intrinsics, which introduces geometric inconsistencies that can hinder model generalization and scalability. To mitigate this issue, we introduce a **Field of View (FoV) unification** strategy. This approach normalizes the input data by projecting all images onto a virtual camera with a predefined, uniform FoV. This process creates a standardized visual input, akin to data captured by a single virtual camera, thereby eliminating intrinsic-related discrepancies for the 3D object detection task. In addition, when creating the instruction data, we mix single-turn and multi-turn formats. The multi-turn format data allows each subsequent box to reference the previous one during training, helping the model learn the layout information.

Furthermore, if we rely solely on template-based 3D object detection data for training, the VLM may overfit to the specific numerical values and fail to generalize spatial understanding. Therefore, to help the VLM better comprehend spatial information at the language level, we introduce the **scene caption**. Unlike general captions, which primarily describe image content, scene captions focus on the layout information and spatial relationships within the image. To obtain such scene captions, we prompt a large VLM (Guo et al., 2025b) with ground-truth 3D bounding boxes and object relationships extracted from the scene graph (Zhu et al., 2023b). The resulting captions not only describe the objects present in the image, but also provide detailed layout information and spatial arrangements.

**Multi-image.** The second category comprises multi-image data, which supports tasks such as multi-view 3D object detection, multi-view correspondence, object-object relationship understanding, and camera motion analysis. These tasks are designed to enhance VLMs in comprehension of spatial relationships across different viewpoints. As illustrated in the third data engine of Figure 3, we sample multi-image data from RGB-D scans sourced from ScanNet (Dai et al., 2017), ScanNet++ (Yeshwanth et al., 2023), and ARKitScenes (Baruch et al., 2021). For correspondence tasks, we utilize point clouds and depth maps from various viewpoints to identify matched points. To unify object information across multiple images, we transform all objects into the camera coordinate system of the first image. For camera motion data, we represent camera poses using Euler angles. After these unification steps, we generate template-based visual instruction samples. In the multi-image scenario, we also create the scene caption to reconstruct the scene layout by text and describe the spatial information represented by multiple RGB images.

**Video.** The third category consists of video data, which enables the model to capture spatiotemporal relationships through tasks such as identifying the order of appearances and counting objects. To construct the video dataset, we employ the same data engine used for multi-image data. The only difference is that we add the appearance time for each object. Furthermore, we enhance the video dataset by sampling two-thirds of the data from VLM-3R (Fan et al., 2025), reorganizing it into a multi-turn format rather than a single-turn format.

With the introduction of the VST-P dataset, the VLM exhibits significantly enhanced fundamental capabilities in comprehending spatial relationships. Notably, there is a $\sim$**20**% improvement on CVBench-3D (Tong et al., 2024), a $\sim$**5**% increase on BLINK (Fu et al., 2024), and a $\sim$**16**% gain on VSIBench (Yang et al., 2025a), as illustrated in Tables 5, 6, and 7.

## 2.2 VST-Reasoning

As shown in Figure 2b, the VST-Reasoning (VST-R) dataset contains 135K samples with two parts: one part includes CoT steps to teach the model how to reason, and the other part provides rule-checkable data used in online RL to improve the reasoning ability. Besides spatial data, both parts include general data to preserve the original general abilities. Most spatial reasoning samples come from multi-image scenarios, which require reconstructing scene details and inferring spatial relations.

For the spatial reasoning samples with the CoT process, we develop a data engine, as illustrated in the bottom left of Figure 3. Specifically, we sample data from template-based question-answer pairs and employ a large VLM (Guo et al., 2025b) as the teacher to generate detailed CoT reasoning steps. Recognizing that the multi-view spatial understanding of the current large VLMs remains limited relative to their general multi-modal abilities, we introduce a novel strategy named **prompting with BEV annotation**. Specifically, this method leverages ground-truth 3D bounding boxes to visualize the BEV image of the scene represented by multiple images. During generation, we provide RGB images, the corresponding BEV visualizations, detailed object information, and question-answer pairs to prompt the teacher VLM. The BEV images serve as an auxiliary spatial prompt, allowing the teacher model to better capture spatial relationships compared to using only RGB images. As a result, the generated reasoning processes are more coherent and accurate. For the CoT patterns, we adopt a textual representation rather than 3D bounding boxes (Ma et al., 2025) or cognition maps (Yin et al., 2025), as the textual format offers greater generality. In particular, during the reasoning process, the model first reconstructs the spatial layout using text, termed RT-CoT, and then infers the answer.

With the VST-R dataset, the VLM demonstrates significantly enhanced spatial reasoning abilities. As illustrated in Table 14, there is an **8.9**% improvement on MMSI-Bench (Yang et al., 2025c).

## 3 Method

Our target is to equip general VLMs with 3D knowledge for better spatial understanding and reasoning from common visual inputs. Therefore, we chose Qwen2.5-VL (Bai et al., 2025) as the base model because it can accurately identify objects and locate them in pixel space. It follows the widely used "ViT-MLP-LLM" paradigm: a pre-trained Vision Transformer (ViT) is combined with a large language model (LLM) via an MLP merger.

### 3.1 Training Strategy

We continued training the base model to endow it with spatial perception and reasoning capabilities. The training process can be divided into three stages.

**Stage 1: Supervised Fine-tuning.** In this stage, we incorporate the foundational spatial understanding into the base model with the proposed VST-P dataset. To maintain the original capabilities of the base model, we also incorporate a portion of general multi-model data from open-source datasets (Li et al., 2024a; Deitke et al., 2025). Assume the base model is parameterized by $\theta$, which can simultaneously process text, images, and video. For any given training sample $x = [x_1, \ldots, x_L]$ of length $L$, we employ visual tokens as the conditioning context for text prediction and adopt the standard auto-regressive objective:

$$\mathcal{L}_\theta(x) = -\sum_{i=2, x_i \in \text{text}}^{L} w_i \log p_\theta(x_i \mid x_1, \ldots, x_{i-1}), \tag{1}$$

**Stage 2: CoT Cold Start.** This stage leverages chain-of-thought (CoT) data to instruct the model utilizing reasoning patterns. For example, in spatial reasoning scenarios with limited viewpoints, the model first reconstructs the layout of the scene using text, and then reasons through the given question. To preserve the model's reasoning ability on general tasks, we also take some general

reasoning data. The training objective remains the same as in Equation 1. The resulting model from this stage has basic spatial reasoning capabilities (Table 9), which serves as the initial RL actor.

**Stage 3: Reinforcement Learning.** In this stage, we employ RL to further enhance the spatial reasoning capabilities of the stage-2 model. For this purpose, we utilize the Group Relative Policy Optimization (GRPO) algorithm (Shao et al., 2024), which bypasses the need for a value model by computing the relative advantage of each response within a group of responses to the same question. To facilitate this process, we curated a verification dataset comprising tasks related to spatial understanding, 3D object detection, and general multi-modal understanding. This dataset is categorized into four task types: multiple-choice, open-ended, OCR, and 3D detection. In the GRPO framework, we employ a mixed rule-based reward to evaluate the generated responses. For a given response $\hat{y}$ and its corresponding ground truth $y$, the overall reward function is defined as:

$$\mathcal{R}(y, \hat{y}) = \mathcal{R}_{\text{acc}}(y, \hat{y}) + \mathcal{R}_{\text{format}}(y, \hat{y}). \tag{2}$$

This function combines an accuracy reward, $\mathcal{R}_{\text{acc}}(\cdot, \cdot)$, which scores the correctness of the response, with a format reward, $\mathcal{R}_{\text{format}}(\cdot, \cdot)$, which incentivizes adherence to a specified output format. For multiple-choice, open-ended, and OCR tasks, the accuracy reward is calculated using standard evaluation protocols (Goyal et al., 2017; Singh et al., 2019; Mathew et al., 2021; Masry et al., 2022). For 3D object detection tasks, the reward is a linear combination of the 3D Intersection over Union (IoU) score and the F1 score:

$$\mathcal{R}_{3d}(y, \hat{y}) = \alpha \mathcal{R}_{\text{iou}}(y, \hat{y}) + (1 - \alpha)\mathcal{R}_{\text{F1}}(y, \hat{y})), \tag{3}$$

where $\alpha$ is a hyperparameter that defaults to $0.5$. In detail, given $N$ predicted and $M$ ground-truth 3D bounding boxes, we first establish a bipartite matching (Kuhn, 1955) between the predictions and the ground truth; $\mathcal{R}_{\text{iou}}(\cdot, \cdot)$ is then calculated as the average IoU of the successfully matched pairs. To calculate $\mathcal{R}_{\text{F1}}(\cdot, \cdot)$, we define a true positive as a match with an IoU score exceeding a threshold of $0.25$. Following this stage, the model exhibits superior spatial reasoning abilities relative to the cold-start model, as shown in Table 14.

### 3.2 Expanding to Vision-Language Action Model

With the spatial-enhanced model, a natural question emerges: can the integration of spatial priors improve the performance of Vision-Language-Action (VLA) models in robotic manipulation tasks? To this end, we adapt the pretrained VLM into a VLA model, following the methodology of OpenVLA (Kim et al., 2024). Specifically, we formulate the action prediction problem as a vision-language task where, given an observation image and a natural language instruction, the model auto-regressively predicts the actions. To accomplish this, we discretize the action space into 256 bins, where each bin corresponds to a special token in the language tokenizer. With the actions tokenized, the entire model is fine-tuned using the objective function defined in Eq 1.

## 4 Experiment

### 4.1 Implementation Details

The training details can be found in Appendix C. For evaluation, we assess spatial understanding across three distinct modalities: single-image ability is benchmarked with CV-Bench (Tong et al., 2024) and 3DSRBench (Ma et al., 2024), multi-image ability with BLINK (Fu et al., 2024) and MMSI-Bench (Yang et al., 2025c), and video-based ability with VSI-Bench (Yang et al., 2025a). The average score (S-AVG) across these benchmarks is used to quantify the overall spatial capabilities. To verify the general ability, we also report the average score (M-AVG) across MMStar (Chen et al., 2024b), MMBench (MMB) (Liu et al., 2024c), RealworldQA (RWQA) (x.ai, 2024), MMMU (Yue et al., 2024), OCRBench (OCRB) (Liu et al., 2024d), and AI2D (Kembhavi et al., 2016). For 3D object detection, we evaluate the model on the SUN RGB-D (Song et al., 2015) (Total3D version (Nie et al., 2020)) and ARKitScenes (Baruch et al., 2021) (Omni3D version (Brazil et al., 2023)).

### 4.2 Main Results

| Methods | CV | 3DSR | MMSI | BLINK | VSI | MMStar | MMB | RWQA | MMMU | OCRB | AI2D |
|---|---|---|---|---|---|---|---|---|---|---|---|
| GPT-4o | 76.0 | 45.3 | 30.3 | 65.9 | 34.0 | 65.1 | 84.3 | 76.2 | 70.7 | 80.6 | 84.9 |
| Gemini-2.5-Pro | - | - | 36.9 | 70.6 | - | 77.5 | 90.1 | 78.0 | 81.7 | 86.6 | 88.4 |
| Seed1.5-VL | 85.2 | 61.6 | 29.7 | 72.1 | 41.5 | 77.8 | 89.9 | 78.4 | 77.9 | 86.1 | 87.3 |
| LLava-OneVision-7B | 61.9 | 54.4 | 26.6 | 48.2 | 32.4 | 61.7 | 80.8 | 66.3 | 48.8 | 62.2 | 81.4 |
| Qwen2.5-VL-3B | 71.8 | 50.2 | 26.5 | 47.6 | 29.6 | 55.9 | 79.9 | 65.4 | 47.9 | 79.7 | 81.6 |
| Qwen2.5-VL-7B | 75.4 | 53.2 | 25.9 | 56.4 | 38.9 | 63.9 | 83.5 | 68.5 | 58.6 | 86.4 | 83.9 |
| InternVL3-8B | 81.0 | 55.7 | 25.7 | 55.5 | 42.1 | 68.2 | 83.4 | 70.8 | 62.7 | 88.0 | 85.2 |
| MiMo-VL-7B-RL | 82.3 | 50.8 | 29.3 | 62.4 | 37.2 | 65.1 | 84.4 | 68.2 | 66.7 | 86.6 | 83.5 |
| SpaceR-7B | 74.8 | 53.3 | 20.1 | 55.4 | 43.5 | 61.6 | 84.3 | 64.7 | 53.1 | 85.9 | 85.5 |
| SPAR-8B | 80.7 | 57.5 | - | 43.9 | 41.1 | - | 79.9 | 64.7 | - | - | - |
| VST-3B-SFT (ours) | 84.4 | 54.1 | 30.2 | 59.1 | 57.9 | 58.0 | 80.9 | 68.4 | 45.2 | 83.7 | 82.5 |
| VST-3B-RL (ours) | 84.2 | 56.5 | 31.3 | 57.2 | 57.7 | 58.9 | 80.5 | 68.5 | 49.8 | 80.9 | 82.4 |
| VST-7B-SFT (ours) | 85.5 | 54.6 | 32.0 | 62.1 | 60.6 | 63.1 | 83.3 | 72.2 | 50.6 | 85.5 | 84.9 |
| VST-7B-RL (ours) | **86.5** | **60.1** | **34.8** | **62.6** | **61.2** | 63.5 | 83.0 | 68.5 | 49.4 | 86.1 | 83.5 |

Table 2: Comparison with state-of-the-art VLMs on spatial benchmarks and general benchmarks.

| Methods | Avg. | Obj. Count | Abs. Dist. | Obj. Size | Room Size | Rel. Dist | Rel. Dir. | Route Plan | Appr. Order |
|---|---|---|---|---|---|---|---|---|---|
| GPT-4o (Achiam et al., 2023) | 34.0 | 46.2 | 5.3 | 43.8 | 38.2 | 37.0 | 41.3 | 31.5 | 28.5 |
| Gemini-1.5-Pro (Team et al., 2024) | 45.4 | 56.2 | 30.9 | 64.1 | 43.6 | 51.3 | 46.3 | 36.0 | 34.6 |
| LLaVA-Video-7B (Zhang et al., 2025b) | 35.6 | 48.5 | 14.0 | 47.8 | 24.2 | 43.5 | 42.4 | 34.0 | 30.6 |
| Qwen2.5-VL-7B (Bai et al., 2025) | 32.7 | 34.5 | 19.4 | 47.6 | 40.8 | 32.8 | 24.5 | 32.5 | 29.4 |
| SAT-7B (Ray et al., 2024) | - | - | - | - | - | 47.3 | 41.1 | 37.1 | 36.1 |
| InternVL-Spatial-8B (Deng et al., 2025) | - | 68.7 | 40.9 | 63.1 | 54.3 | 47.7 | - | 29.9 | 60.5 |
| SpaceR-7B (Ouyang et al., 2025) | 43.5 | 61.9 | 28.6 | 60.9 | 35.2 | 38.2 | 46.0 | 31.4 | 45.6 |
| VILASR-7B (Wu et al., 2025) | 45.4 | 63.5 | 34.4 | 60.6 | 30.9 | 48.9 | 45.2 | 30.4 | 49.2 |
| VLM-3R-7B (Fan et al., 2025) | 60.9 | 70.2 | **49.4** | 69.2 | 67.1 | **65.4** | **80.5** | **45.4** | 40.1 |
| VST-3B-SFT (ours) | 57.9 | 69.3 | 45.4 | 71.8 | 62.4 | 59.0 | 46.0 | 38.7 | **70.2** |
| VST-3B-RL (ours) | 57.7 | 66.6 | 45.0 | 72.8 | 60.9 | 59.9 | 47.6 | 40.7 | 68.3 |
| VST-7B-SFT (ours) | 60.6 | **72.0** | 44.4 | 74.3 | 68.3 | 59.7 | 55.8 | 44.9 | 65.2 |
| VST-7B-RL (ours) | **61.2** | 71.6 | 43.8 | **75.5** | **69.2** | 60.0 | 55.6 | 44.3 | 69.2 |

Table 3: Detailed comparison with state-of-the-art VLMs on VSI-Bench (Yang et al., 2025a).

| Methods | $AP_{15}$ |
|---|---|
| Seed1.5-VL | 33.5 |
| Gemini-2.0-Pro | 32.5 |
| Gemini Robotics-ER | **48.3** |
| Implicit3D | 24.1 |
| Total3DU | 14.3 |
| VST-3B-SFT (ours) | 37.3 |
| VST-3B-RL (ours) | 40.1 |
| VST-7B-SFT (ours) | 41.6 |
| VST-7B-RL (ours) | **44.2** |

Table 4: Comparison $AP_{15}$ on SUN RGB-D.

As shown in Table 2, our VST models achieve competitive results across both spatial and general benchmarks. Notably, VST-7B-SFT and VST-7B-RL deliver leading performance on mainstream spatial understanding tasks. On the CV-Bench (Team, 2024), VST-7B-SFT attains 85.5, surpassing the proprietary Seed1.5-VL (Guo et al., 2025b) with 85.2. On MMSI-Bench (Yang et al., 2025c), VST-7B-SFT achieves 32.0, outperforming GPT-4o (Achiam et al., 2023) at 30.3, while RL further boosts VST-7B-RL to 34.8, approaching the proprietary state of the art Gemini-2.5-Pro (Comanici et al., 2025) at 36.9. Notably, the VSI-Bench (Yang et al., 2025a) highlights the strength of our models in video spatial understanding. VST-7B-SFT reaches 60.6, and VST-3B-SFT achieves 57.9, substantially ahead of GPT-4o with 34.0. Detailed results are reported in Table 3. Without any specialized 3D encoder, VST-7B-RL delivers the best overall average among comparable VLMs, achieving 61.2. Although VLM-3R-7B (Fan et al., 2025) attains a similar score, it relies on an additional expert 3D encoder, whereas VST operates with a standard vision backbone. Beyond the overall average, VST shows clear strengths on fine-grained spatial sub-tasks: it leads in Object Size and Room Size estimation, and performs strongly in Relative Direction and Appearance Order. In addition, VST models provide a well-balanced performance on general benchmarks. These outcomes highlight the clear advantage of VST in spatial perception and reasoning while maintaining strong competitiveness in multi-modal understanding. Table 4 summarizes results on the SUN RGB-D (Song et al., 2015) 3D object detection benchmark. VST-7B-SFT reaches 41.6 $AP_{15}$, while VST-7B-RL improves to 44.2, ranking first among both general VLMs (Comanici et al., 2025; Guo et al., 2025b) and expert methods (Zhang et al., 2021; Nie et al., 2020). These findings show that VST, even without auxiliary 3D encoders, can achieve strong 3D object detection purely through visual spatial tuning.

## 4.3 ABLATION STUDY

**Ablation for the single-image data.** Our baseline model is the Qwen2.5-VL-3B fine-tuned on a general dataset of 800K samples. From this baseline, we incrementally introduced different types of data to enhance its capabilities. As presented in Table 5, the incorporation of this data yields a 20.8%

| Data | S-AVG | Single-image | | | Multi-image | | Video | M-AVG |
|---|---|---|---|---|---|---|---|---|
| | | CV-2D | CV-3D | 3DSR | MMSI | BLINK | VSI | |
| Baseline | 49.9 | 71.2 | 72.6 | 50.5 | 26.1 | 49.2 | 29.6 | 68.3 |
| + 3Dod | 50.9 | 71.5 | 78.3 | 51.0 | 25.5 | 48.2 | 30.9 | 70.0 |
| + 3D Grounding | 50.5 | 73.3 | 72.3 | 50.5 | 27.7 | 48.4 | 31.0 | 69.5 |
| + Scene Caption (*si*) | 52.8 | 72.4 | 83.8 | 52.2 | 25.9 | 49.4 | 33.1 | 69.8 |
| + Measurement (*si*) | 53.3 | 71.9 | 83.0 | 53.0 | 26.7 | 49.4 | 35.5 | 69.2 |
| + Depth and Distance Data | **56.4** | 73.5 | 93.4 | 53.2 | 28.8 | 50.6 | 38.7 | 69.7 |

Table 5: Ablation for the single-image data. *si* denotes single-image data.

| Data | S-AVG | Single-image | | | Multi-image | | Video | M-AVG |
|---|---|---|---|---|---|---|---|---|
| | | CV-2D | CV-3D | 3DSR | MMSI | BLINK | VSI | |
| Baseline | 56.4 | 73.5 | 93.4 | 53.2 | 28.8 | 50.6 | 38.7 | 69.7 |
| + Corespondence | 56.7 | 74.1 | 92.5 | 53.9 | 27.9 | 52.4 | 39.1 | 69.7 |
| + 3Dod (*mi*) | 56.5 | 73.3 | 92.3 | 53.1 | 30.0 | 52.7 | 37.7 | 69.2 |
| + Object-object Relation | 57.1 | 74.3 | 91.9 | 53.2 | 31.9 | 53.2 | 38.3 | 69.4 |
| + Camera-camera Relation | 57.3 | 72.7 | 93.1 | 53.6 | 31.8 | 53.7 | 38.8 | 68.8 |
| + Scene Caption (*mi*) | 57.4 | 73.2 | 92.8 | 54.0 | 32.1 | 53.0 | 39.3 | 69.2 |
| + Camera Motion | **57.7** | 73.9 | 92.5 | 54.0 | 32.4 | 55.0 | 38.2 | 68.7 |
| + General Data (*mi*) | **57.7** | 73.8 | 92.4 | 53.7 | 32.7 | 55.4 | 38.4 | 69.2 |

Table 6: Ablation for the multi-image data. *mi* denotes mingle-image data.

| Data | S-AVG | Single-image | | | Multi-image | | Video | M-AVG |
|---|---|---|---|---|---|---|---|---|
| | | CV-2D | CV-3D | 3DSR | MMSI | BLINK | VSI | |
| Baseline | 54.8 | 73.8 | 92.4 | 53.7 | 32.7 | 55.4 | 38.4 | 69.2 |
| + General Video | 57.9 | 74.3 | 92.3 | 53.5 | 31.9 | 57.6 | 38.1 | 69.4 |
| + VST-video | **60.6** | 75.1 | 93.1 | 54.0 | 31.3 | 55.6 | 54.7 | 69.4 |

Table 7: Ablation for the video data.

improvement on CV-Bench-3D (Tong et al., 2024). Notably, the inclusion of single-image data also enhances performance on multi-image and video benchmarks, with improvements of $+2.7\%$ on MMSI-Bench (Yang et al., 2025c) and $+9.1\%$ on VSI-Bench (Yang et al., 2025a).

**Ablation for the multi-image data.** Based on the ability to infer spatial information from single images, we further enhance the model by integrating multi-image data to capture spatial relationships across diverse viewpoints. As shown in Table 6, the multi-image data results in a 1.3% increase in the average spatial understanding score. Notably, this approach yields a 3.9% improvement on MMSI-Bench (Yang et al., 2025c) and a 4.8% improvement on BLINK (Fu et al., 2024).

**Ablation for the video data.** Since single-image and multi-image data provide minimal improvement on video benchmarks, we further constructed video data to enhance spatial understanding for video inputs. As shown in Table 7, general video data from LLaVA-OneVision (Li et al., 2024a) exhibits limitations in video-based spatial understanding tasks. By incorporating our VST-video data, the model achieves a 16.6% improvement on VSI-Bench (Yang et al., 2025a).

**3D Object Detection Settings.** We present ablation studies on 3D object detection settings in Table 8. First, unifying FoV (Uni-FoV) across different datasets yields a 2.5 AP improvement on ARKitScenes. Second, substituting Euler angles with quaternions results in reduced performance. Third, replacing all multi-turn data with an equivalent amount of single-turn (ST) data leads to a decrease of 1.7 AP on SUN RGB-D and 1.8 AP on ARKitScenes.

| Settings | SUN RGB-D | | | | | ARKitScenes | | | | |
|---|---|---|---|---|---|---|---|---|---|---|
| | AP | $AP_{15}$ | $AP_{25}$ | $AP_{50}$ | $AP_{100}$ | AP | $AP_{15}$ | $AP_{25}$ | $AP_{50}$ | $AP_{100}$ |
| Baseline | 18.5 | 28.5 | 19.1 | **3.2** | **30.7** | 29.4 | 42.7 | 31.8 | 7.2 | **44.6** |
| w/o Uni-FoV | **18.6** | **29.1** | **19.2** | 2.2 | 30.4 | 26.9 | 40.9 | 28.2 | 5.4 | 42.0 |
| Quat | 18.3 | 29.4 | 17.8 | 2.8 | 30.6 | 28.6 | 42.4 | 30.8 | 6.3 | 43.9 |
| ST Data | 16.8 | 26.0 | 17.0 | 2.7 | 28.4 | 27.6 | 41.9 | 29.0 | 6.0 | 42.5 |

Table 8: Ablation for different settings for 3D object detection. Training data: 600K 3D object detection samples and 800K general samples.

**Ablation for the spatial reasoning data.** We conduct an ablation study on CoT data for spatial reasoning, as summarized in Table 9. Our baseline is Qwen2.5-VL-7B (Yang et al., 2024a), fine-tuned on one-third of VST-P single-image data, multi-view correspondence data, multi-image 3D object detection data, and 800K general samples. First, we let the model represent the layout of scenes using 3D bounding boxes (Num-CoT), achieving a score of 29.2 on MMSI-Bench. However, estimating camera poses across diverse viewpoints proves challenging. Thus, we propose reconstructing the scene using text within the CoT (RT-CoT), which outperforms Num-CoT. Furthermore, RT-CoT from prompting with BEV annotations (RT-CoT$_{BEV}$) yields an additional 1.1% improvement. Finally, mixing all types of data raises the score to 31.7%. RL further improves the score to 35.3% (Table 14).

| CoT Type | Data | Overall | Positional Relationship | | | | | | Attribute | | Motion | | MSR |
|---|---|---|---|---|---|---|---|---|---|---|---|---|---|
| | | | CC | OO | RR | CO | OR | CR | M | A | C | O | - |
| - | - | 26.4 | 22.6 | 28.7 | 17.3 | 39.5 | 38.8 | 28.9 | 20.3 | 18.2 | 21.6 | 28.9 | 25.3 |
| Num-CoT | OO | 29.2 | 28.0 | 33.0 | 28.4 | 40.7 | 25.9 | 36.1 | 39.1 | 36.4 | 20.3 | 31.6 | 18.7 |
| RT-CoT | OO | 30.0 | 35.5 | 31.9 | 25.9 | 44.2 | 36.5 | 32.5 | 32.8 | 30.3 | 21.6 | 27.6 | 21.2 |
| RT-CoT$_{BEV}$ | OO | 31.1 | 31.2 | 36.2 | 27.2 | 46.5 | 40.0 | 36.1 | 26.6 | 30.3 | 17.6 | 30.3 | 24.8 |
| RT-CoT$_{BEV}$ | Mix | **31.7** | 35.5 | 39.4 | 37.0 | 44.2 | 36.5 | 43.4 | 28.1 | 22.7 | 17.6 | 30.3 | 21.7 |

Table 9: Cold-start results for spatial reasoning on the MMSI-Bench (Yang et al., 2025c). Data: OO refers to the object-object subset, while Mix includes all data types.

| Accuracy Reward | AP | AP$_{15}$ | AP$_{25}$ | AP$_{50}$ | AP$_{100}$ |
|---|---|---|---|---|---|
| Baseline | 20.2 | 30.3 | 20.6 | 4.5 | 33.5 |
| 3D IoU + Recall | 13.8 | 20.9 | 14.7 | 2.6 | 33.6 |
| 3D IoU + F1 Score | **24.4** | **36.0** | **26.3** | **5.8** | **37.2** |

Table 10: Ablation for accuracy reward $\mathcal{R}_{3d}(\cdot, \cdot)$ on SUN RGB-D

| VLA Backbone | Spatial | Object | Goal | 10 | Avg. |
|---|---|---|---|---|---|
| Qwen2.5-VL-3B | 56.6 | 86.6 | 53.8 | 15.2 | 53.1 |
| Qwen2.5-VL-3B (VST) | **65.0** | **88.4** | **67.8** | **25.6** | **61.7** |

Table 11: Success rate comparison on the LIBERO benchmark (Liu et al., 2023). Qwen2.5-VL-3B (VST) refers to the model fine-tuned on our VST dataset.

**Ablation for the accuracy reward on 3D object detection task.** As recorded in Table 10, when we use 3D IoU and recall as the accuracy reward, performance drops markedly because each ground-truth box is matched with too many false-positive predictions. Thus, we switch to IoU and F1 score as the accuracy reward, which yields a 4.2 AP improvement.

## 4.4 EXPANDING TO VLA MODEL

As detailed in Section 3.2, we adapt our VST-tuned VLM into a VLA model. In contrast to the approach used by OpenVLA (Kim et al., 2024), we do not utilize any pre-trained data on robotic learning. Instead, we directly fine-tune the VLM and its action embeddings on the small-scale LIBERO benchmark (Liu et al., 2023) from scratch for action prediction. The results are presented in Table 11. Notably, the VLA model based on VST-3B, which incorporates spatial knowledge, surpasses the one based on Qwen2.5-VL-3B (Bai et al., 2025) by an average of $8.6\%$ in success rate. This improvement clearly demonstrates that the integration of spatial knowledge provides a significant performance benefit to VLA models.

## 5 RELATED WORK

VLMs have become a cornerstone of modern AI. Recent advancements on VLMs have focused on several areas: employing more powerful vision encoders (Chen et al., 2024d), supporting dynamic resolutions (Wang et al., 2024b), incorporating reasonable positional embeddings (Bai et al., 2025), and curating higher-quality data (Li et al., 2024a). Despite these significant improvements, established benchmarks (Yang et al., 2025a;c) reveal a persistent deficiency in the spatial understanding and reasoning capabilities of current VLMs. Early attempts introduced specialized datasets generated by expert models (Chen et al., 2024a), while SAT (Ray et al., 2024) tackled data scarcity by simulation. Subsequent works (Daxberger et al., 2025; Zhang et al., 2025a) have further expanded data to enhance these spatial abilities. There are also efforts (Wu et al., 2024; Ouyang et al., 2025; Yin et al., 2025) to improve spatial reasoning. In contrast to previous approaches that treat spatial understanding and reasoning as distinct tasks, our proposed method integrates foundational spatial perception and reasoning capabilities, aiming to achieve a more holistic and robust model for spatial intelligence.

## 6 CONCLUSION

We present Visual Spatial Tuning (VST), a general and scalable framework that endows vision-language models with human-like spatial perception and reasoning abilities. With the large-scale perception data (VST-P) and curated reasoning instructions (VST-R), VST effectively acquires spatial awareness without degrading general capabilities. The proposed approach achieves state-of-the-art performance on multiple spatial benchmarks, demonstrating that spatial abilities in foundation models can be systematically scaled. Moreover, the Vision-Language-Action (VLA) models are proved to be enhanced with better visuo-spatial skills, enabling more grounded interaction with the physical world. The generality, scalability, and effectiveness of VST highlight a promising direction toward building physical AI systems that reason and act in space with human-like intelligence.

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

## A  DATA CONVENTION

### A.1  CAMERA COORDINATE

In our work, we define the camera coordinate system based on the right-hand rule. The camera center is taken as the origin, with the X-axis pointing to the right (parallel to the image plane), the Y-axis pointing downward (also parallel to the image plane), and the Z-axis pointing forward along the optical axis. Within this system, a 3D bounding box is specified by its center coordinates $(x, y, z)$, its size $(x_l, y_l, z_l)$, and its orientation $(pitch, yaw, roll)$. In our convention, the X-dimension corresponds to the front–back size of the box, Y represents the vertical extent, and Z denotes the lateral (side) extent. In our definition of a 3D bounding box, the X-axis points to the front, the Y-axis points downward, and the Z-axis points sideways. The rotation angles are defined as the transformations from the camera axes to the box axes, measured in degrees and normalized by $\pi$ (i.e., divided by $180°$). Distances and dimensions are given in meters.

### A.2  INSTRUCTION FORMAT

---

**The CoT Format**

< |im_start| >system
You are a helpful assistant. You should first think about the reasoning process in the mind and then provide the user with the answer. The reasoning process is enclosed within < think > < /think > tags, i.e. < think > reasoning process here ¡/think¿ answer here.< |im_end| >
< |im_start| >user
< |vision_start| >image.jpg< |vision_end| >**{question}**< |im_end| >
< |im_start| >assistant
< think >**{thinking content}**< /think > **{answer}**< |im_end| >

---

## B  EXTENDED RELATED WORK

**Large Vision-Language Models.** Recently, Large Vision–Language Models (LVLMs) have become a pivotal technology in artificial intelligence, able to understand and integrate information across multiple modalities such as text, images, and video. Most LVLMs use an Encoder–MLP–LLM architecture (Liu et al., 2024b; Li et al., 2024a; Wang et al., 2024b; Bai et al., 2025; Deitke et al., 2025; Chen et al., 2024c). Specifically, a pre-trained vision encoder—often the CLIP vision encoder (Radford et al., 2021)—extracts visual embeddings, and a projector (MLP layers) projects these embeddings into the language embedding space. Researchers have improved this approach in several ways: using more advanced vision encoders (Zhai et al., 2023; Chen et al., 2024d), increasing input resolution (Liu et al., 2024a; Chen et al., 2024c), adopting dynamic resolution (Wang et al., 2024b), refining multimodal positional embeddings (Yang et al., 2024a; Bai et al., 2025), and synthesizing high-quality training data (Li et al., 2024a; Cho et al., 2025). Beyond the Encoder–MLP–LLM architecture, several works (Zhu et al., 2023a; Dai et al., 2023; Ye et al., 2023; Li et al., 2024b) employ a Q-former (Li et al., 2023) to map visual embeddings into the language space. The Q-former replaces long visual feature sequences with a fixed set of learnable queries, reducing the length and complexity of the visual input. Flamingo (Alayrac et al., 2022) integrates gated cross-attention layers into LLMs to facilitate cross-modal alignment, while CogVLM (Wang et al., 2023) adds specialized vision experts to each LLM block to align visual and language features. Additionally, some works (Bavishi et al., 2023; Yang et al., 2025b) aim to develop LVLMs as unified models that consolidate vision and language capabilities. Additionally, some studies (Huang et al., 2025; Meng et al., 2025; Xu et al., 2025c) explore the use of reinforcement learning in the post-training stage to enhance the visual reasoning capabilities of LVLMs, an approach inspired mainly by DeepSeek-R1 (Guo et al., 2025a). The majority of these works apply Generalized Reinforcement Learning from Preference Optimization (GRPO) (Shao et al., 2024) to train LVLMs, achieving significant improvements in many tasks.

**Large Vision-Language Models for Spatial Understanding and Reasoning.** Despite the remarkable progress of current LVLMs in visual tasks (Yue et al., 2024; Liu et al., 2024c;d; Mathew et al., 2021; Xie et al., 2024), numerous benchmarks (Yang et al., 2025a;c; Ma et al., 2024) have

highlighted persistent challenges in spatial understanding and reasoning. To address these issues, SpatialVLM (Chen et al., 2024a) pioneered the application of VLMs to spatial understanding by constructing VQA datasets using expert models. Similarly, SpatialRGPT (Cheng et al., 2024) expanded RGB-based spatial understanding to the RGB-D domain by generating spatial datasets from 3D scene graphs. Recognizing the prohibitive cost of collecting and annotating real-world data, SAT (Ray et al., 2024) employed simulators to generate training data, thereby extending its focus from static to dynamic tasks. SpatialBot (Cai et al., 2024) enables VLMs to invoke external tools for depth estimation, thereby improving their ability to interpret spatial information in input images. Subsequent studies (Daxberger et al., 2025; Zhang et al., 2025a; Xu et al., 2025a; Deng et al., 2025) have further advanced the field by constructing more comprehensive datasets to enhance the spatial understanding capabilities of VLMs. In parallel, another line of research focuses on enhancing the spatial reasoning abilities of VLMs. For example, MVoT (Wu et al., 2024) leverages multimodal representations within its reasoning traces to strengthen spatial reasoning. SpaceR (Ouyang et al., 2025) and MindCube (Yin et al., 2025) incorporate textual cognition maps into their reasoning traces to enhance spatial reasoning, further improving performance through reinforcement learning. Similarly, Spatialreasoner (Ma et al., 2025) performs spatial reasoning by predicting 3D locations and poses as intermediate results. VILASR (Wu et al., 2025) enhances spatial reasoning by incorporating visual tools and introducing visual prompting into the reasoning process. In contrast to these prior studies, which typically focus exclusively on either spatial understanding or spatial reasoning, our approach begins with foundational capabilities and builds upon them to enhance the model's overall reasoning skills.

## C  MORE IMPLEMENTATION DETAILS

**Stage 1.** The initial training stage aims to establish a strong foundation of spatial understanding capabilities. For this stage, we use a global batch size of $128$, a sequence length of $16,384$, and a dynamic data packing strategy to accelerate the training process. We employ the AdamW (Loshchilov & Hutter, 2019) optimizer, setting the base learning rate to $5 \times 10^{-5}$ and the vision encoder's learning rate to $5 \times 10^{-6}$. During this phase, we combine our VST data with general multi-modal data from LLaVA-OneVision (Li et al., 2024a). This approach allows the model to learn new spatial understanding knowledge while mitigating catastrophic forgetting of its original capabilities. For our ablation studies, we use Qwen2.5-VL-3B (Bai et al., 2025) as the base model, training it on a mixture of one-third of the VST data and $800K$ general multi-modal samples. For our final models, we employ Qwen2.5-VL-3B, Qwen2.5-VL-7B, and Qwen2.5-VL-32B as base models, utilizing the entire VST dataset combined with $2.4M$ general multi-modal samples.

**Stage 2.** In the CoT cold-start stage, we continue training the model from the initial foundation stage. The hyper-parameters are adjusted to a global batch size of $64$, a base learning rate of $1 \times 10^{-5}$, a vision encoder learning rate of $1 \times 10^{-6}$, and a sequence length of $16,384$. In this stage, the training data is a mixture of spatial reasoning data and general multimodal reasoning data. We train the model for 2 epochs, as we observed that smaller-scale models require extended training to effectively master the long-form CoT reasoning process.

**Stage 3.** In the RL stage, we further refine the model from the second stage using the VeRL (Sheng et al., 2025) framework. For the training objective, we adopt a revised version of the GRPO algorithm (Yu et al., 2025). This stage utilizes the AdamW (Loshchilov & Hutter, 2019) optimizer with a constant learning rate of $1 \times 10^{-6}$ and a global batch size of $128$.

**VLA Model.** For the expansion to the VLA model, we continued to use the AdamW (Loshchilov & Hutter, 2019) optimizer, but with a modified learning rate schedule: a base learning rate of $8 \times 10^{-5}$ and a vision encoder learning rate of $8 \times 10^{-6}$. We set the global batch size to $128$ and the max sequence length for data packing to $2048$. This adjustment was necessary to compensate for the relatively short action sequences and the small resolution of the training images ($256 \times 256$). The model is finetuned for 50 epochs on the LIBERO dataset (Liu et al., 2023) in total.

## D  MORE EXPERIMENTS

**Data scaling of spatial foundational tasks.** To enable the VLM to perceive the positions of objects in 3D space, we selected monocular 3D object detection and depth estimation as our foundational

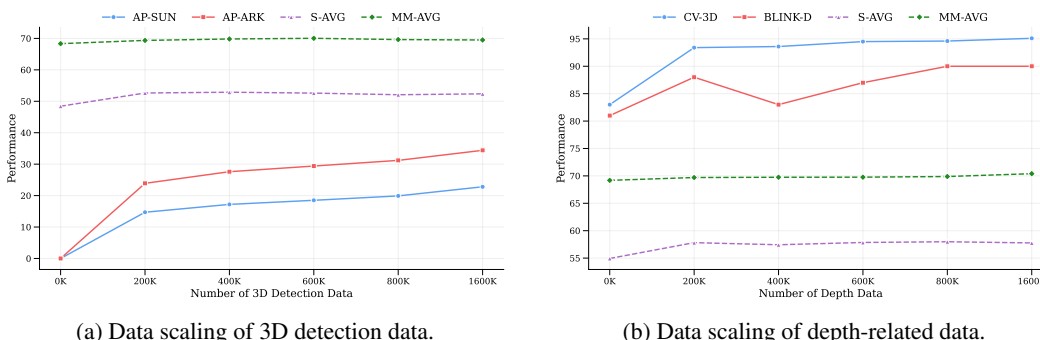

(a) Data scaling of 3D detection data.      (b) Data scaling of depth-related data.

Figure 4: Data scaling of spatial foundational tasks.

tasks. We then incrementally scaled the volume of training data to validate the VLM's emerging spatial perception capabilities. As shown in Figure 4a, the AP on the SUN-RGBD (Song et al., 2015) and ARKitScenes (Baruch et al., 2021) datasets progressively improved as the amount of 3D detection data increased, demonstrating its ability to learn how to perceive the 3D spatial positions of objects from visual input. Furthermore, by gradually introducing depth-related data, we discovered that the VLM could learn to judge the relative distances between objects and the camera, even with a comparatively small amount of data, as illustrated in Figure 4b.

**Scaling Model Size.** We further investigate the relationship between model size and spatial understanding performance, as shown in Table 12. Increasing the model size from 3B to 7B results in a 1.3% improvement in average scores on spatial understanding benchmarks. Further increasing the model size to 32B yields a 1.7% improvement in average scores on these benchmarks. These results suggest that larger models achieve greater improvements on spatial understanding tasks.

The relationship between model size and 3D object detection is presented in Table 13. When increasing the model size from 3B to 7B, we observe a 4.2 AP improvement on SUN-RGBD (Song et al., 2015) and a 3.5 AP improvement on ARKitScenes (Baruch et al., 2021). However, when increasing the model size from 7B to 32B, the performance does not exhibit a positive correlation as seen in the spatial understanding benchmarks. This may be because a model with 7B parameters is already sufficient to handle this fundamental perception task.

**Scaling Data.** When the dataset was increased threefold, all models showed improvement. As shown in Table 12, tripling the data scale resulted in a 1.1% improvement for the 3B model, a 1.5% improvement for the 7B model, and a 0.7% improvement for the 32B model.

| Model Size | Data Scale | S-AVG | Single-image | | | Multi-image | | Video | M-AVG |
|---|---|---|---|---|---|---|---|---|---|
| | | | CV-2D | CV-3D | 3DSR | MMSI | BLINK | VSI | |
| 3B | 1× | 60.6 | 75.1 | 93.1 | 54.0 | 31.3 | 55.6 | 54.7 | 69.4 |
| | 3× | 61.7 | 75.0 | 93.8 | 54.1 | 30.2 | 59.1 | 57.9 | 69.9 |
| 7B | 1× | 61.9 | 76.5 | 94.4 | 53.5 | 31.9 | 58.3 | 57.0 | 73.1 |
| | 3× | 63.4 | 75.2 | 95.7 | 54.6 | 32.0 | 62.1 | 60.6 | 73.3 |
| 32B | 1× | 63.6 | 77.5 | 93.1 | 55.7 | 36.9 | 61.4 | 57.1 | 75.6 |
| | 3× | 65.3 | 80.7 | 94.8 | 56.1 | 36.0 | 65.4 | 58.7 | 75.9 |

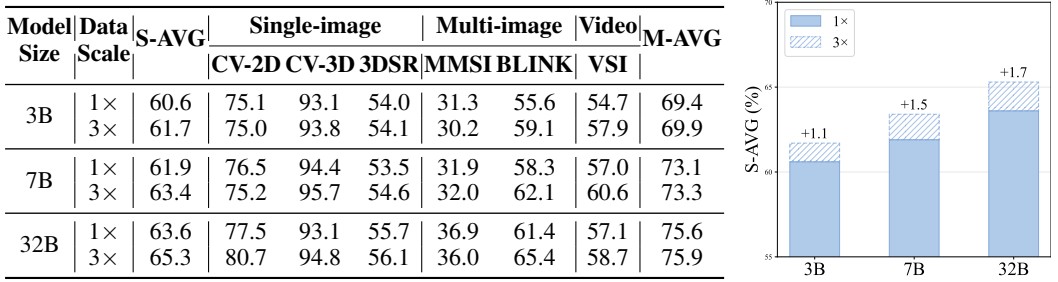

Table 12: Results of scaling model and data size

**Spatial reasoning with RL.** We found that after the CoT cold start, the model performs better without CoT inference than with it. As shown in Table 14, the model achieves 33.6% accuracy on MMSI-Bench (Yang et al., 2025c) without CoT inference, surpassing the CoT setting by 1.9%. This suggests the model learns spatial knowledge from the CoT process, but its CoT reasoning ability is weak, leading to poorer results. Therefore, to improve the spatial reasoning ability, we apply

| Model Size | Data Scale | SUN-RGBD | | | | | ARKitScenes | | | | |
|---|---|---|---|---|---|---|---|---|---|---|---|
| | | AP | AP₁₅ | AP₂₅ | AP₅₀ | AP₁₀₀ | AP | AP₁₅ | AP₂₅ | AP₅₀ | AP₁₀₀ |
| 3B | 1× | 20.2 | 30.3 | 20.6 | 4.5 | 33.5 | 31.5 | 45.1 | 34.8 | 8.3 | 46.6 |
| | 3× | 26.5 | 37.3 | 28.9 | 7.1 | 39.7 | 38.0 | 51.7 | 41.5 | 14.3 | 53.4 |
| 7B | 1× | 24.2 | 37.4 | 24.6 | 5.4 | 37.7 | 35.0 | 48.8 | 38.0 | 11.3 | 50.1 |
| | 3× | 28.0 | 41.6 | 29.5 | 7.7 | 42.1 | 39.1 | 52.8 | 42.3 | 15.3 | 54.3 |
| 32B | 1× | 19.6 | 29.5 | 19.5 | 4.3 | 32.7 | 31.1 | 44.3 | 33.7 | 8.9 | 46.3 |
| | 3× | 22.5 | 33.2 | 23.3 | 5.1 | 36.1 | 33.6 | 47.6 | 35.8 | 11.3 | 49.1 |

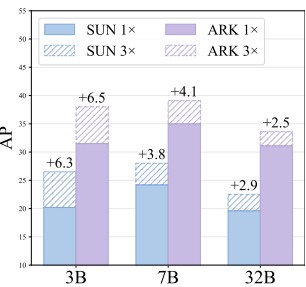

Table 13: 3D object detection results of scaling model and data size

online RL (Shao et al., 2024) as detailed in Section 3.1. After RL, the model reaches 35.4% on MMSI-Bench (Yang et al., 2025c) when using CoT inference.

| Training Stage | CoT Inference | Overall | Positional relationship | | | | | | Attribute | | Motion | | MSR |
|---|---|---|---|---|---|---|---|---|---|---|---|---|---|
| | | | CC | OO | RR | CO | OR | CR | M | A | C | O | - |
| - | ✗ | 26.4 | 22.6 | 28.7 | 17.3 | 39.5 | 38.8 | 28.9 | 20.3 | 18.2 | 21.6 | 28.9 | 25.3 |
| Cold start | ✗ | 33.6 | 43.0 | 35.1 | 33.3 | 43.0 | 36.5 | 44.6 | 32.8 | 27.3 | 28.4 | 32.9 | 23.2 |
| Cold start | ✓ | 31.7 | 35.5 | 39.4 | 37.0 | 44.2 | 36.5 | 43.4 | 28.1 | 22.7 | 17.6 | 30.3 | 21.7 |
| RL | ✗ | 34.7 | 34.4 | 38.3 | 29.6 | 53.5 | 37.6 | 43.4 | 25.0 | 36.4 | 27.0 | 40.8 | 25.3 |
| RL | ✓ | 35.3 | 35.5 | 38.3 | 30.9 | 54.6 | 35.3 | 47.0 | 26.6 | 48.5 | 33.8 | 31.6 | 22.7 |

Table 14: RL results for spatial reasoning on the MMSI-Bench (Yang et al., 2025c).

# E    MORE DATASET DETAILS

## E.1    PROMPTING WITH BEV ANNOTATION

As shown in Table 15, we prompt the Seed1.5-VL (Guo et al., 2025b) with the BEV visualization to generate the CoT process.

## E.2    DATASET VISUALIZATION

As shown in Table 16, 17, 18, 19, 20, 21, and 22, we provide a visualization of sample data from various tasks within the VST-P dataset.

**Prompting with BEV annotation**

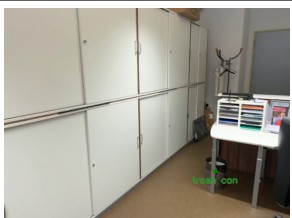 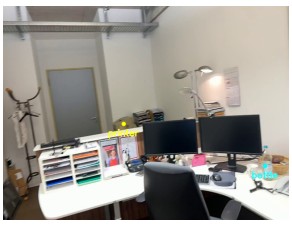 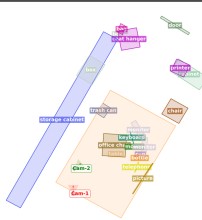

| | |
|---|---|
| **Prompt** | \<system prompt\> \<ground-truth\> \<question\>
If the small metal trash can on the floor near the white storage cabinets in the first image is north of the small printer on the white cabinet near the door in the second image, what direction is the clear plastic bottle on the white desk with two monitors in the second image from the printer? Options: A. southeast, B. northeast, C. south, D. northwest |
| **Teacher model** | First, I look at the first image. The scene has a long row of white storage cabinets along the left wall, a white desk with shelves holding folders on the right, and a small metal trash can sitting on the floor between the cabinets and the desk. There's also a coat hanger in the corner and some boxes stacked nearby. The trash can is clearly positioned on the left side of the room, close to the storage cabinets.
Next, I examine the second image, which was taken from a closer, more forward angle-like the camera moved toward the desk and tilted up slightly. The desk is now in the foreground, holding two black monitors, a keyboard, a mouse, a telephone, and a clear plastic bottle. Against the back wall near the door, there's a small printer on a white cabinet. The door and coat hanger from the first image are still visible in the background, but the trash can isn't present here.
Putting both images together, the room's layout makes sense: the left wall has storage cabinets, the center/right has a desk, and the back wall has a door. The trash can (from image 1) is on the left side near the cabinets, the printer (from image 2) is on the back right near the door, and the bottle (from image 2) is on the foreground desk.
The question says the trash can is north of the printer. Since the trash can is on the left side of the room and the printer is on the back right, "north" from the printer's perspective must be toward the left-toward the trash can. Now, I need to figure out where the bottle is relative to the printer. The bottle is on the desk, which is in the foreground of the second image. From the printer's position (back right near the door), the desk is in front of it-closer to the camera. The bottle is on the part of the desk that's to the left of the printer (since the desk extends from the foreground to the back wall, and the printer is on a cabinet against the back wall to the right of the desk).
So, from the printer's view: the bottle is to the left (the same direction as north, since north is toward the trash can on the left) and forward (since the desk is closer than the trash can). Combining these, the bottle is in the northwest direction from the printer. That matches option D. |

Table 15: One example to illustrate the prompting with BEV annotation.

**Scene Caption**

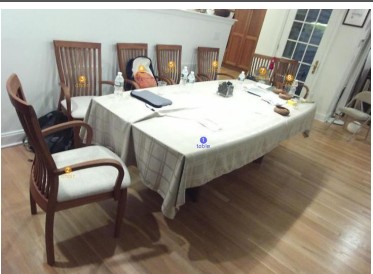

| | |
|---|---|
| **Prompt** | Analyze the spatial organization and positional relationship in this image. |
| **Answer** | Starting from the camera's viewpoint, the nearest object is a chair (orange point-2) with a depth of 1.72m. Adjacent to it, behind, is another chair (orange point-3) at 2.72m depth. Moving forward, a table (blue point-1) spans 3.00m depth. To the table's left, chairs (orange points-4,5,6) are aligned at depths 3.20m, 3.63m, and 4.11m respectively. On the table's right, chairs (orange points-7,8) are aligned, with chair (orange point-7) beside chair (orange point-8). Chair (orange point-6) is in front of chair (orange point-8). Distances: chair (orange point-2) is 1.79m from the table; chair (orange point-3) is 1.58m from the table. Chairs (orange points-3,4,5,6) form a left-aligned row. Chairs (orange points-6,7,8) form a right-aligned row. Chair (orange point-8) is 0.52m from chair (orange point-7). |

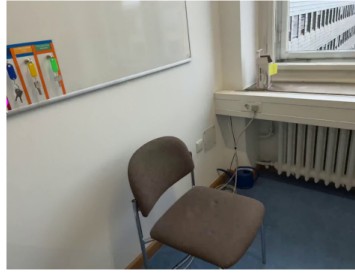 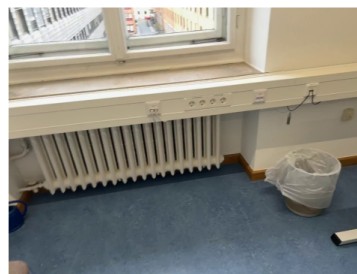

| | |
|---|---|
| **Prompt** | The observer's line of sight is preserved as images are captured one by one. Describe these two frames in detail. |
| **Answer** | The first image shows a room scene with a whiteboard on the left wall, a chair in the lower central area, a window on the right side with a view of a building outside, a heater below the window, a box on the windowsill, sockets on the wall near the heater, and a light switch on the wall. Key holders with colorful keys are attached to the whiteboard. The second image, taken with the camera rotated right from the first perspective, shows part of the window and the heater still visible. New elements include a trash can on the floor, a table partially in view, and additional sockets. Objects present in the first image such as the whiteboard, light switch, and the box on the windowsill are no longer in view, while the chair remains present. |

Table 16: Example of scene caption

**Depth-related Data**

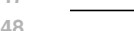

| | |
|---|---|
| **Prompt** | Tell me the depth relationship of the objects of A.sink, B.shelves, C.blinds, D.lamp. |
| **Answer** | The objects from near to far is A, D, B, C |

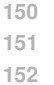

| | |
|---|---|
| **Prompt** | There are several boxes in the image: box-A, box-B. Each box represents an object. Present the object represented by these boxes in an order that goes from close to far and give their names. |
| **Answer** | box-A (chair), box-B (picture) |



| | |
|---|---|
| **Prompt** | You are given several 2D bounding boxes in the image: [3, 132, 98, 468] [322, 83, 475, 481] Arrange the object represented by these boxes from the farthest to the nearest based on their depth relationship and give their names. Output the sorted bboxes and labels using JSON format. |
| **Answer** | |

```
[
        {"bbox_2d":[324,81,477,475],"label":"person"},
        {"bbox_2d":[3,130,98,462],"label":"person"}
]
```

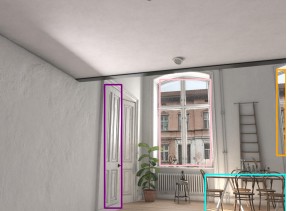

| | |
|---|---|
| **Prompt** | After reviewing the image, which object—table (cyan box), window-A (orange box), window-B (pink box) and door (purple box)—would you say is nearer to the camera? |
| **Answer** | table |

Table 17: Example of depth-related data.

**Distance-related Data**

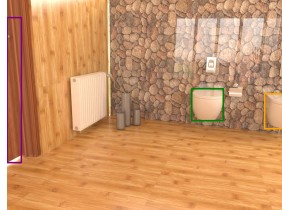

| Prompt | Considering the positions, which object—the toilet-A (orange box) or the toilet-B (green box)—do you think is farthest from the door (purple box)? |
|---|---|
| **Answer** | toilet-A |

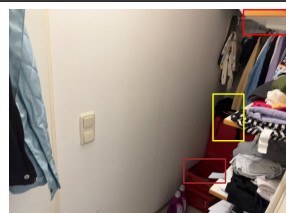

| Prompt | Given the spatial layout, which object among the shelf (red box) or the basket (brown box) is situated farthest from the bag (yellow box)? |
|---|---|
| **Answer** | shelf |

Table 18: Example of distance-related data.

**Measurement-related Data**

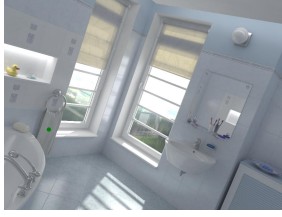

| Prompt | Can you specify the height of the blinds pointed out by the blue point? State the measurement in centimeters. |
|---|---|
| **Answer** | 103cm |

Table 19: Example of measurement-related data.

**Correspondence Data**

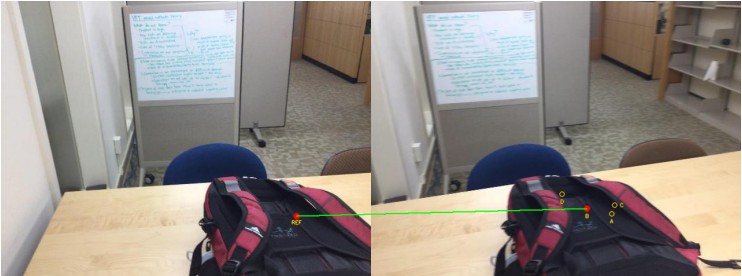

| | |
|---|---|
| **Prompt** | The first image shows a point circled in gold. After adjusting the camera or lighting, the second image presents several gold-circled points labeled 'A, B, C, D'. Which matches the original? Options: A: point-A, B: point-B, C: point-C, D: point-D |
| **Answer** | B: point-B |

Table 20: Example of correspondence data.

**Camera-related Data**

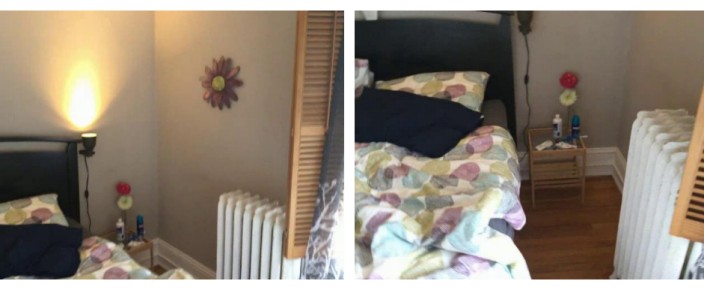

| | |
|---|---|
| **Prompt** | The frames are captured in a continuous manner from a first-person perspective. Which way is the camera's perspective moving? Options: A. moving backward, B. moving rightward and forward, C. moving backward and upward, D. moving leftward |
| **Answer** | B. moving rightward and forward |

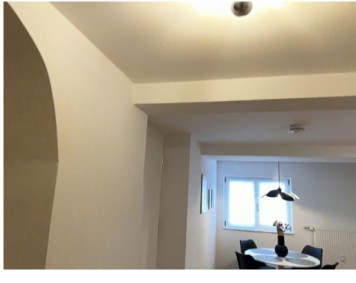 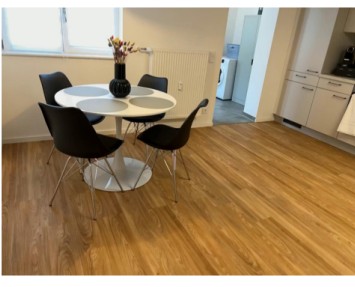

| | |
|---|---|
| **Prompt** | The frames are acquired in a continuous sequence from a first-person perspective. If the first picture was taken with the camera facing west, what is the direction for the second picture? Options: A. southeast, B. north, C. south, D. northwest |
| **Answer** | D. northwest |

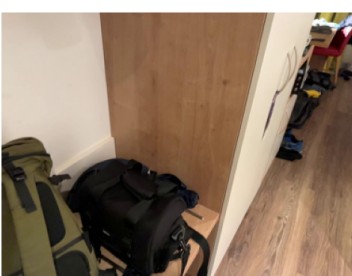 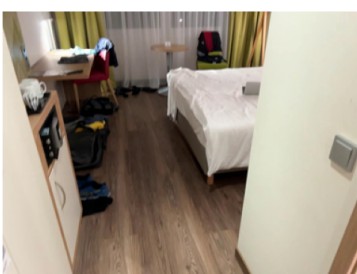

| | |
|---|---|
| **Prompt** | Images are shot one after another from a first-person perspective. When positioned at the second photo spot, how is the first camera placed relative to me? Options: A. right, B. back, C. front, D. front right |
| **Answer** | B. back |

Table 21: Example of camera-related data.

**Object-related Data**

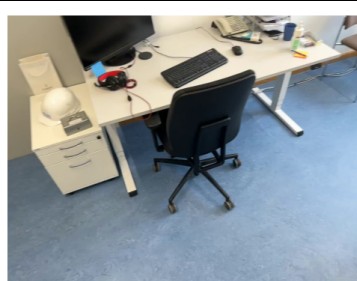 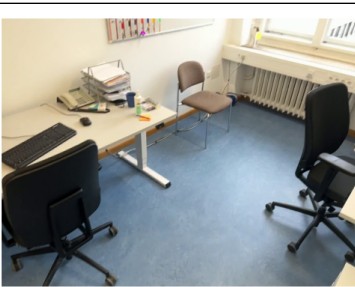

| | |
|---|---|
| **Prompt** | If the small white cabinet under the white desk is north of the black monitor on the left side of the desk, what direction is the chair on the right side of the room from the black monitor? Options: A. southeast B. north C. south D. southwest |
| **Answer** | A. southeast |

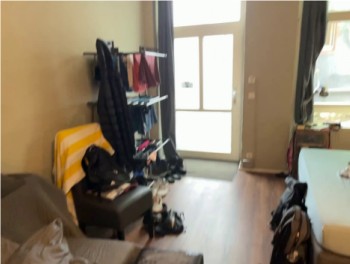 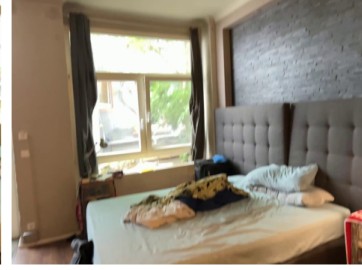

| | |
|---|---|
| **Prompt** | If, from the camera position of the first image, the direction toward the hanging jacket (visible in the first image) is north, then in which direction does the window (visible in the second image) lie relative to the first image's camera? Options: A. northeast, B. southeast, C. east, D. south |
| **Answer** | A. northeast |

Table 22: Example of object-related data.

