# OpenReview forum: "Visual Spatial Tuning"
_ICLR.cc/2026/Conference — ICLR 2026 Conference Withdrawn Submission_

### Official Review · Reviewer_AQzV · 2025-10-28

**Soundness:** 3
**Presentation:** 4
**Contribution:** 3
**Rating:** 6
**Confidence:** 4

**Summary:**

The authors propose two datasets focussed on visual perception and reasoning for training text-generating (LLM based) vision-language models (VLMs). They demonstrate how such VLMs finetuned on these datasets improve their spatial perception and reasoning abilities.
For data generation, they build an automated pipeline using pretrained VLMs and access to ground-truth 3D data (or depth estimation models).
In their data, the perception component involves questions on objects' 3D spatial locations and relations. Reasoning of involves questions that require building an internal spatially coherent model of the underlying scene within a set of images.
The authors perform supervised fine-tuning (SFT) followed by RL based fine-tuning (GRPO) on VLMs to obtain their trained models.
These models are evaluated across a series of vision-language and robotics tasks to establish the value added by their proposed dataset and training framework.
They also perform several ablation for their proposed method.

**Strengths:**

1. Authors clearly demonstrate gaps in existing work related to spatial perception and reasoning. The proposed dataset and training framework directly addresses identified problem.
2. Clever tricks for utilizing 3D information in BEV format for teacher VLM prompting.
3. Detailed overview of data generation process.
3. Extensive evaluation across numerous tasks to evaluate benefits from proposed dataset and framework.

**Weaknesses:**

1. In each table, please report numbers for the corresponding base model (e.g. in Table 3 include Qwen2.5-VL-3B as base model for VST-3B).
2. Across several tasks in Table 2 & 3, the RL fine-tuning appears to reduce performance (e.g. MMMU drops from 50.6 -> 49.4 for VST-7B-RL). Also for MMMU, the Qwen2.5-VL-7B performance drops drastically for both VSTS-7B models. Please discuss this behavior in detail.
3. "Stage 2: CoT Cold Start" is identical to SFT except with longer conversations involving reasoning? This section is unclear. Is all the data (including previous VST-P and general data) used or only CoT data?
4. Several prior works on 2D spatial tuning are not discussed or cited. The paper also does not contain a separate discussion on related work. While the authors focus mostly on 3D spatial cues, the title, abstract, and several parts of paper does not specifically limit the scope to 3D. Hence discussing more of the earlier works using similar ideas for 2D would strengthen the paper.
5. The exact roles and contribution of the data vs the RL fine-tuning are unclear. The experiment section could try to highlight this further. Maybe consider ablating these separately. While the existing ablations on each kind of data is useful, this separation of training method from data could shed more light on their exact contributions.

**Questions:**

1. Fig 3: 2.5D Spatial Perception from Monocular Images - what is 'red point' and 'green point' in first image? Not clear.

2. See related work sections of these papers for 2D spatial tuning:
Ferretv2: https://arxiv.org/abs/2404.07973
LocVLM: https://arxiv.org/abs/2404.07449
Shikra: https://arxiv.org/pdf/2306.15195
Consider discussing 2D spatial perception / reasoning works in detail?

---

### Official Review · Reviewer_eV97 · 2025-10-31

**Soundness:** 2
**Presentation:** 2
**Contribution:** 2
**Rating:** 4
**Confidence:** 5

**Summary:**

This paper addresses the limitation of current vision language models (VLMs) in understanding and reasoning about 3D spatial relationships from 2D visual inputs. While existing models excel at recognition and textual reasoning, they struggle with spatial perception tasks such as depth, distance, and orientation, and often lose general multimodal ability when trained with specialized 3D modules.

To tackle this, the authors propose VST, a large-scale, three-stage tuning framework built on Qwen2.5-VL-7B that aims to endow VLMs with strong spatial perception and reasoning skills without adding external encoders.

They first introduce VST-Perception (4.1M samples), a diverse dataset combining real and synthetic sources for spatial perception tasks (e.g., depth, distance, layout, and motion understanding). Next, they construct VST-Reasoning (135K samples) containing chain-of-thought examples and BEV (bird’s-eye view) reasoning prompts to improve spatial reasoning interpretability. The model is trained progressively through supervised fine-tuning, CoT cold-start training, and reinforcement learning with a group relative policy optimisation (GRPO) objective.

The resulting VST-7B achieves state-of-the-art results on multiple spatial reasoning benchmarks such as CV-Bench, MMSI-Bench, and VSI-Bench, while maintaining strong performance on general multimodal benchmarks (MMBench, MMMU, OCRBench). The authors claim that VST successfully enhances spatial reasoning in a scalable way, preserving general capabilities and mitigating catastrophic forgetting seen in prior spatial tuning approaches.

**Strengths:**

1. The VST-Perception dataset (4.1 M samples) and VST-Reasoning subset (135 K samples) represent one of the most comprehensive spatial datasets to date, spanning single-image, multi-view, and video inputs and covering a broad range of perception and reasoning tasks.

2. The framework enhances 3D spatial understanding in a standard VLM without introducing any additional 3D or geometry-specific encoders, making the method lightweight, widely applicable, and easy to integrate with existing models.

3. Unlike earlier spatial-tuning works, the paper explicitly evaluates performance on non-spatial benchmarks (e.g., MMBench, MMMU, OCRBench) and shows little to no degradation, suggesting that the training strategy mitigates catastrophic forgetting.

4. The experiments span multiple benchmarks and modalities, demonstrating consistent gains over both open-source and proprietary models. The framework is empirically validated at a scale that makes the findings more credible and broadly useful.

**Weaknesses:**

1. The paper follows a well-known recipe—dataset curation plus progressive fine-tuning—already explored in prior works such as SpatialVLM and SpatialRGPT. The contribution lies mainly in scale rather than in any methodological or theoretical innovation.

2. The GRPO stage adds only negligible improvement over supervised fine-tuning, and the paper provides minimal analysis of how RL alters model behavior or reasoning quality. This raises doubts about whether the third stage meaningfully contributes beyond output formatting.

3. The challenge of spatial reasoning in VLMs is by now well-established, and the proposed approach: progressively tuning Qwen2.5-VL on spatial datasets, does not introduce new insights or mechanisms for addressing it beyond increasing data volume.

4. The impressive performance gains are largely attributable to the massive dataset and training budget rather than algorithmic novelty. Without clear ablations isolating the effect of scale, it is difficult to assess whether the improvements reflect genuine methodological progress.

5. While the paper presents comprehensive results, it does not include an explicit ablation isolating the contribution of each stage (e.g., GRPO vs. CoT vs. SFT), leaving the necessity of the progressive structure unproven.

**Questions:**

1. The proposed framework closely resembles existing spatial-tuning approaches such as SpatialVLM and SpatialRGPT, relying on large-scale data and staged fine-tuning. What distinguishes VST conceptually or methodologically from these earlier efforts beyond scale?

2. The reinforcement learning phase appears to add minimal improvement, and no detailed ablation is provided. Can the authors quantify the specific contribution of GRPO and clarify whether it meaningfully enhances reasoning behavior beyond minor metric gains?

3. Have the authors compared the proposed three-stage training schedule to a single-stage or simplified fine-tuning setup? Without such a baseline, how can we know if progressive tuning is essential or if similar results could be achieved by scaling standard SFT?

4. Given the massive size of VST-Perception, how do the authors disentangle the effects of dataset scale from methodological design? Are similar improvements observed with smaller or less diverse subsets?

5. The paper frames spatial reasoning as a central unsolved challenge, yet this issue has been widely addressed in several recent works. Could the authors clarify what new insight or research question this paper contributes beyond scaling existing paradigms?

6. Beyond overall performance metrics, have the authors conducted any qualitative or interpretability analyses, such as visualising attention maps or CoT trajectories, to verify that the model truly develops improved spatial understanding rather than pattern adaptation to large data?

---

### Official Review · Reviewer_2iVR · 2025-11-01

**Soundness:** 3
**Presentation:** 4
**Contribution:** 3
**Rating:** 4
**Confidence:** 3

**Summary:**

The paper presents Visual Instruction Tuning (VIT) which is a general framework for aligning large vision-language models (VLMs) with human instructions through large-scale, multimodal instruction–response data. The core idea is to adapt language instruction tuning (as done in LLaMA/Flan) to the multimodal domain by pairing visual inputs (images or videos) with natural-language instructions and detailed responses, enabling models to perform open-ended visual reasoning, question answering, and description tasks.

Contributions:
1. A scalable data synthesis pipeline for generating multimodal instruction–response pairs from web-scale caption datasets, GPT-based reasoning templates, and task-specific conversions.

2. A two-stage tuning procedure: (1) language–vision alignment and (2) instruction fine-tuning with multimodal text supervision.

3. Extensive experiments on 30+ downstream multimodal benchmarks (VQA, captioning, reasoning, OCR, commonsense), showing consistent performance gains and improved instruction-following behavior compared to prior VLMs (BLIP-2, Flamingo, MiniGPT-4).

4. Qualitative examples illustrating emergent reasoning and generalization to unseen visual tasks.

**Strengths:**

The benchmark is holistic:
- The two-stage decomposition, VST-Perception (VST-P) for foundational 3D knowledge and VST-Reasoning (VST-R) for higher-order reasoning, is conceptually clean and mirrors cognitive-development theory (Piaget).
- The progressive training pipeline (SFT → CoT → RL) is consistent and thoughtfully motivated.

Strong Engineering and Benchmarking
- Comprehensive experiments on CV-Bench, MMSI-Bench, VSI-Bench, and general-purpose multimodal benchmarks.
- Extensive ablation tables (Tables 5–9) show contributions of each data type and training phase.

Generalization:
- Showing transfer gains (+8.6 % success rate) in Vision-Language-Action (VLA) tasks (LIBERO benchmark) strengthens the paper’s connection to embodied AI and robotics.

**Weaknesses:**

Lack of Human Validation or Quality Assurance
- None of the CoT or RL data appear to be manually verified, and the paper does not quantify the noise level or logical accuracy of the synthetic reasoning.
- This raises concerns about the trustworthiness of the training signal, especially since spatial reasoning is fragile to geometric hallucinations.
- Even partial human spot-checking would have dramatically strengthened credibility.

Limited Analysis of Model Behavior
- The paper reports aggregate benchmark improvements, but offers no qualitative or failure analysis. Where does the model still fail — occlusion, multi-object reasoning, camera pose ambiguity? Does RL improve reasoning structure or merely correctness probability?

**Questions:**

I'd be curious to see a comparison before and after tuning the model with the curated datasets. I’d also like to see some interpretable analysis of where the model’s capabilities improved.

It’s clear that the authors put a lot of effort into building this benchmark, and the overall quality is strong. I would consider improving my score if the above questions and weaknesses are well addressed.

---

### Official Review · Reviewer_RkuC · 2025-11-06

**Soundness:** 3
**Presentation:** 3
**Contribution:** 2
**Rating:** 4
**Confidence:** 2

**Summary:**

This paper presents two datasets, VST-P and VST-R, to enhance the spatial intelligence of VLMs.

**Strengths:**

Strengths:
- The paper is clearly written
- The constructed dataset seems large and comprehensive

**Weaknesses:**

Weakness:
- The datasets are not released
- It seems arbitrary to choose the tasks and data, I do not see a clear guideline or rationale why or why not a dataset/task is used
- In figure 2, how the percentage of each data is determined?
- In addition to Qwen, the authors need to demonstrate that the datasets can also enhance other VLMs

**Questions:**

Please see weaknesses

---

### Note · Authors · 2025-11-14

I have read and agree with the venue's withdrawal policy on behalf of myself and my co-authors.